

# Transcriptomic and proteomic profiles of II YOU 838 (*Oryza sativa*) provide insights into heat stress tolerance in hybrid rice

Yan Wang[1,2], Yang Yu[1], Min Huang[2], Peng Gao[2], Hao Chen[2], Mianxue Liu[2], Qian Chen[2], Zhirong Yang[1] and Qun Sun[1]

[1] Key Laboratory of Bio-resource and Bio-environment of the Ministry of Education, College of Life Science, Sichuan University, Chengdu, Sichuan, China
[2] Key Laboratory of Irradiation Preservation of Sichuan Province, Sichuan Institute of Atomic Energy, Chengdu, Sichuan, China

## ABSTRACT

Heat stress is an increasing threat to rice production worldwide. To investigate the mechanisms of heat tolerance in hybrid rice and their contributions to rice heterosis, we compared the transcriptome of the hybrid rice II YOU 838 (II8) with the transcriptomes of its parents Fu Hui 838 (F8) and II-32A (II3) after heat stress at 42 °C for 0 h, 24 h, 72 h and 120 h. We also performed a proteomic analysis in II8 after heat stress at 42 °C for 24 h. The transcriptome data revealed time-dependent gene expression patterns under the heat stress conditions, and the heat stress response of II8 was greatly different from those of its parents. Gene ontology analysis of the differentially expressed genes that were clustered using *k*-means clustering showed that most of the up-regulated genes were involved in responses to stimuli, cell communication, and metabolic and transcription factor activities, whereas the down-regulated genes were enriched in photosynthesis and signal transduction. Moreover, 35 unique differentially abundant proteins, including a basic helix-loop-helix transcription factor (bHLH96), calmodulin-binding transcription activator, heat shock protein (Hsp70), and chaperonin 60 (CPN60), were detected in the proteomic analysis of II8 under heat stress. The co-regulatory analysis revealed novel genes and pathways involved in heat tolerance, namely, ferredoxin-NADP reductase, peroxidases, mitogen-activated protein kinase kinase kinase, and heat shock factor (HSF)–Hsp network. Members of the Hsp and HSF families had over-dominant expression patterns in the hybrid compared with its parents, to help maintain the higher photosynthesis and antioxidant defense systems in the hybrid. Our study suggests that the complex HSF–Hsp regulatory network contribute to the heat tolerance of the hybrid rice.

# INTRODUCTION

Global warming has led to an increase in extreme weather events in recent decades (*Wang et al., 2019*). Extreme weather conditions, such as high temperature and drought, have greatly affected rice production and food security worldwide. Cultivated rice

Corresponding author
Qun Sun, qunsun@scu.edu.cn

(*Oryza sativa*) is the staple food for more than half of the world's population (*Zhang et al., 2012b*). *O. sativa* subsp. *indica* accounts for more than 70% of rice production globally (*Zhang et al., 2016b*). It is, however, very sensitive to high-temperature stress during almost all growth stages, particularly the flowering period (*Wang et al., 2019*; *Zafar et al., 2018*). Rice grain yields decline by 10% for each 1 °C increase in the minimum night time temperature during the growing period (*Peng et al., 2004*). Heat stress can cause irreversible damage by retarding plant growth, metabolic activities, spikelet fertility, and seed setting, thus reducing the rice production (*Wang et al., 2019*). Heat stress also may significantly suppress the photosynthetic rate, hormone levels, membrane stability, respiration, and primary and secondary metabolites (*Zafar et al., 2018*). At the cell level, heat shock induces thermo-tolerance responses and activates the expression of genes encoding heat shock proteins (Hsps) and other related proteins, which promotes the synthesis of Hsps, antioxidants, and other osmoprotectants to offset heat stress-induced biochemical and physiological changes (*Hasanuzzaman et al., 2013*). Consequently, understanding the molecular basis of heat tolerance will greatly contribute to the development of new strategies for improving heat tolerance in rice (*Cheabu et al., 2018*; *Li et al., 2018b*).

The tolerance of rice to heat stress is linked with the ability to perceive the high temperature stimulus and generate and transmit the signal, as well as with the expression of specific genes and production of metabolites, and the initiation of the antioxidative defense system (*Anjum et al., 2016*; *Cao et al., 2008*; *Zafar et al., 2018*). In response to sudden heat shock, a number of mechanisms and several compounds are produced to protect rice plants from stress. Heat stress damages the internal cellular environment, activates the heat tolerance machinery, and triggers the expression of heat shock genes. This initiates the production of Hsps, antioxidant enzymes, free radical scavengers, and osmoprotectants to detoxify reactive oxygen species (ROS) and recover the cell membrane so that cellular homeostasis is reestablished (*Hasanuzzaman et al., 2013*; *Li et al., 2018a*; *Zhao et al., 2018*). In transgenic plants over expressing an Hsp gene, ROS scavenging and antioxidant enzyme activity increased under heat stress (*Liu et al., 2018*). The heat tolerance of rice is associated with complex regulatory networks, and the genes involved in the related pathways may respond differently at different time points in different tissues (*Hasanuzzaman et al., 2013*). RNA sequencing (RNA-Seq) is a powerful high-throughput technique that has been used to systematically investigate the molecular reactions by which rice responds and adapts to different heat stress environments (*Sarkar, Kim & Grover, 2014*; *Zhang et al., 2012b*, *2013*). Proteomic analyses of rice under heat stress conditions have been reported (*Damaris et al., 2016*; *Gammulla et al., 2010*; *Zou, Liu & Chen, 2011*). However, there are currently no published studies on the dynamic changes at the proteome and transcriptome levels during a prolonged heat stress regime in rice flag leaves, especially during the flowering stage. Although a heat shock response is a short-term reaction, heat stress usually lasts for a long time over all rice developmental stages in the field. Previous studies focused mainly on changes at the beginning of stress (*Gammulla et al., 2010*; *Sarkar, Kim & Grover, 2014*; *Zhang et al., 2012b*; *Zou, Liu & Chen, 2011*).

In this study, a relatively longer heat shock response time was studied to obtain a more comprehensive understanding of the regulatory network involved in heat stress tolerance in rice.

Rice hybrids usually show strong heterosis, which is a complex multigenic trait that can be extrapolated as the sum total of physiological and phenotypic traits, including resistance to biotic and abiotic environmental stresses (*Zhang et al., 2016b*). To overcome the challenges presented by global warming, it is important to understand how rice hybrids perceive and respond to high temperatures. Hybrid rice II YOU 838 (II8), which was obtained from a cross between Fu Hui 838 (F8) and II-32A (II3), exhibits superior multiple agronomic traits including yield and adaptability, especially resistance to high temperatures. Thus, II8 has been cultivated widely in China (*Su et al., 2013*; *Wang et al., 2016b*). Although its resistance to heat stress has been confirmed in the field, the mechanism remains to be deciphered. Although many studies have been conducted on heat stress mechanisms using heat resistant materials and mutants, they are different from rice varieties such as II8 that are used in production (*Han et al., 2018*; *Poli et al., 2013*; *Wang et al., 2016a*; *Zhang et al., 2018*).

We compared the transcriptomic data of flag leaves from the hybrid II8 variety with the transcriptomic data of flag leaves from its parents during long-term heat stress by RNA-Seq. We also surveyed the mechanisms underlying the higher heat tolerance of II8 by transcriptomic and proteomic analyses. F8 is a mutation-containing variety derived from *O. sativa* subsp. *indica* MH63 (*Deng et al., 2009*). The *indica* varieties have high levels of genetic diversity; therefore, we used the MH63 genome and proteome data as the references in this study (*Song et al., 2018*; *Zhang et al., 2016a*, *2016b*). The combination of transcriptomic and proteomic profiles provided insights into the mechanisms of heat tolerance in hybrid rice and their contributions to heterosis.

# MATERIALS AND METHODS

## Plant materials and heat treatment

The hybrid rice line II8, along with the paternal F8 and maternal II3 lines, were cultivated in the experimental field in Sichuan Province, China. The field management followed essentially normal agricultural practice. At the heading stage (108 days after sowing), the plants were transferred to a growth chamber maintained at 22 °C/32 °C (13 h/11 h dark/light cycle) with 80% humidity and light intensity of 600 μmol m$^{-2}$ s$^{-1}$ for 48 h to allow the plants to adapt to the growth chamber. Then, the control flag-leaf samples (0 h) were collected from the II8, F8, and II3 lines at 19:00 (i.e., after 11 h light at 32 °C and before 13 h dark at 32 °C) and frozen immediately in liquid nitrogen. On the first flowering day, the II8, F8, and II3 lines were heat treated at 32 °C/42 °C (13 h/11 h dark/light cycle) for 120 h during the blossoming days. The flag leaves of the three lines were harvested at 24 h, 72 h and 120 h, and frozen immediately in liquid nitrogen. Light, humidity, water, and fertilizer conditions in the growth chamber were all the same during the 168 h experimental period. Each sample had three biological replicates during the flowering stage.

## Total RNA and total protein isolation

Total RNA was extracted from the flag leaves of the three lines using TRIzol (Invitrogen, Carlsbad, CA, USA) and treated with DNase I (Thermo Fisher Scientific, Waltham, MA, USA) following the manufacturer's instructions. The purity of the RNA was assessed by determining the absorbance levels at 260 nm and 280 nm using a spectrophotometer. RNA integrity was checked using an Agilent 2100 Bioanalyzer (Agilent, Santa Clara, CA, USA). The RNA samples from three independent biological replications were mixed equally and used for RNA-Seq. The same RNA samples from three biological replicates were used independently for quantitative real-time RT-PCR (qPCR).

Total proteins was extracted from the flag leaves of II8 plants that were subjected to the same heat stress conditions at 0 h and 24 h using a two-step precipitation/extraction method (*Jiang et al., 2012*). Three independent biological replicates were used.

## Library sequencing and data processing

The sequencing libraries of II8 and the parental lines were constructed using digital gene expression profiling techniques as described in a previous study (*Tao et al., 2012*). The RNA-Seq was carried out on an Illumina HiSeq 2,000 platform (Beijing Genomics Institution, Shenzhen, China). The transcript levels were quantified using the transcripts per million (TPM) methods (*Zhang et al., 2012a*). The same RNA samples from II8 flag leaves also were analyzed using a deeper RNA-Seq method. Briefly, the RNA-Seq libraries were constructed as described in a previous study (*Xing et al., 2017*).
The expression level of each gene was normalized to fragments per kilobase per million (*Trapnell et al., 2010*). The Minghui 63 (MH63) reference genome sequence (GenBank accession: LNNK00000000) was downloaded from the Rice Information Gateway (http://rice.hzau.edu.cn) (*Song et al., 2018*; *Zhang et al., 2016b*). A fold change *Dias et al. (2011)* with an absolute value of log2 FC > 1 and a stringent false discovery rate <0.001 were set as the thresholds to define differentially expressed genes (DEGs). A GO enrichment analysis was performed using the singular enrichment analysis in agriGO (*Tian et al., 2017*) with *indica* as the reference background (http://bioinfo.cau.edu.cn/agriGO/index.php). Hypergeometric tests were performed using the default parameters and Benjamini–Hochberg corrected *p*-values. Hierarchical clustering, *k*-means clustering, and the STRING database (v10.5) (*Szklarczyk et al., 2017*) analysis were performed to determine co-regulatory patterns. The network was visualized in Cytoscape (*Baryshnikova, 2016*), with genes as nodes and interactions as links (edges) between the nodes. The RNA-Seq data have been deposited in the NCBI Sequence Read Archive (https://www.ncbi.nlm.nih.gov/sra) under accession number SRP168528.

## qPCR analysis

The qPCR analysis was carried out using the SYBR Green 2.0 × PCR reaction mix (TaKaRa, Japan). The primer pairs were designed using NCBI Primer-Blast (*Ye et al., 2012*) and synthesized by Invitrogen (Table S1). The relative expression levels of all the candidate genes were normalized against the expression levels of three internal controls, rice ACTIN1 gene MH03g0618400 (*Jain et al., 2006*), MH01g0139700, and
MH02g0177100, as described previously (*Vandesompele et al., 2002*). For each gene, three technical replicates and three independent biological replicates were used at each sampling time point.

## Two-dimensional polyacrylamide gel electrophoresis (2D-PAGE) and data analysis

The 2D-PAGE was performed as described by *Jiang et al. (2012)*. After electrophoresis, the gels containing the three biological replicates were silver stained (*Sinha et al., 2001*) and scanned at an optical resolution of 300 dpi using a UMAX scanner (UTA-1100, GE Healthcare, Chicago, IL, USA). The digitalized images were analyzed using ImageMaster 2D Platinum software (v5.0, GE Healthcare, Chicago, IL, USA). The spot volume was normalized as a percentage of the total volume of all the spots in a gel (vol%). Spot abundance (vol%) ratios >2 with $p < 0.05$ were set as the thresholds to identify differentially abundant proteins (DAPs). Protein spots of interest were excised from the gels, digested, and then subjected to nanoelectrospray ionization followed by tandem mass spectrometry (MS/MS) in an LTQ Orbitrap Velos (Thermo Fisher Scientific, San Jose, CA, USA) coupled online to high-performance liquid chromatography instrument. Proteins were identified by searching the MH63 reference proteome, which was downloaded from the Rice Information Gateway using the Mascot peptide search engine v2.3.02 (Matrix Science, London, UK).

## RESULTS

Three *indica* rice varieties, II8 and the parental lines F8 and II3, were heat treated for 120 h. Visible damage to the flag leaves was detected after 120 h of heat stress (Fig. 1A). Under the same heat stress conditions, the damage to II8 flag leaves was the least serious, followed by F8 flag leaves; the damage to II3 flag leaves was the most obvious. This result indicated that II8 was more tolerant to the high temperature than its parents. In a previous study, physiological and biochemical adaptations, such as chlorophyl content, and peroxidase, superoxide dismutase, and catalase activity, were higher in II8 flag leaf than in the flag leaves of the paternal lines under heat-stress conditions (*Wang et al., 2016b*). The differences that we observed in the flag leaf phenotypes confirmed the appropriateness of using II8 and the parental lines for the transcriptome analysis.

## Differential gene expression patterns in II8 and the parental lines in response to different durations of heat stress

The DEGs were detected using the thresholds FC ≥ 2-fold and false discovery rate <0.001. In the three comparisons (0 h vs 24 h, 0 h vs 72 h, 0h vs 120 h), more DEGs was up-regulated than down-regulated DEGs in all three lines (Fig. 1B). This suggests that under the heat stress conditions, more genes were switched on than switched off to promote thermo-tolerance. A total of 4,016, 3,073, and 3,596 DEGs were detected in II8, F8, and II3, respectively. The higher number of DEGs that were found in the hybrid may be related to the higher heat resistance of II8 compared with its parents. We consider that

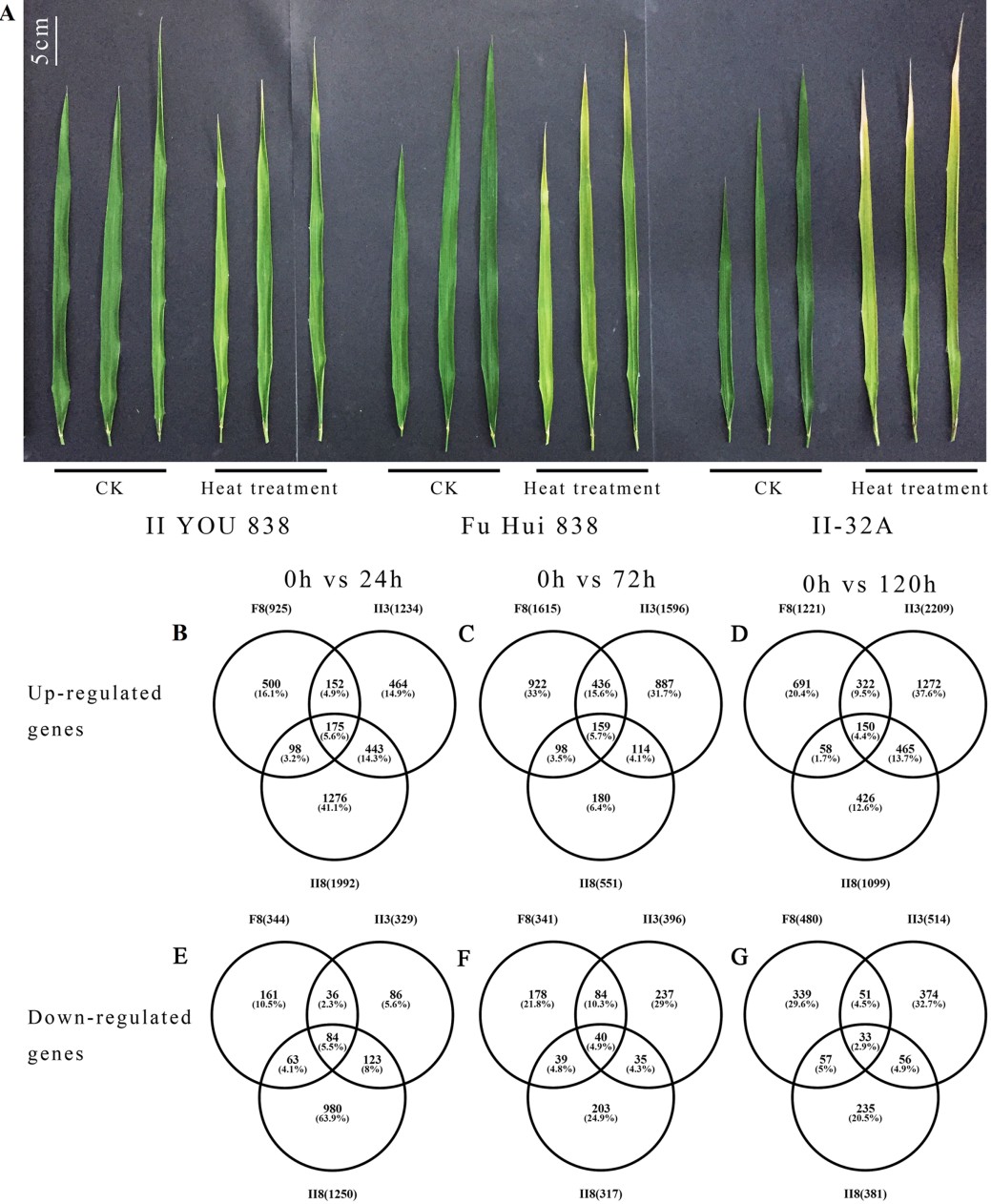

**Figure 1 Changes in flag leaves and gene expression between hybrid rice (II8) and the paternal F8 and maternal II3 lines after heat stress.** (A) Flag leaves of II8 and the parental lines after 120 h of heat stress at 42 °C. CK, unheated (0 h) control group; heat treatment, heat treated for 120 h group. (B–D) Numbers of the up-regulated differentially expressed genes between II8 and the parental lines after 24 h, 72 h, 120 h heat stress. (E–G) Numbers of the down-regulated differentially expressed genes between II8 and the parental lines after 24 h, 72 h, 120 h heat stress. II8, hybrid rice II YOU 838; F8, paternal Fu Hui 838; II3, maternal II-32A.

changes in the transcriptome are mainly the results of long-term adaptive events associated with heat stress tolerance.

The total number of DEGs (3,242) and exclusive number of DEGs (1,274 up-regulated and 980 down-regulated) detected in the hybrid after 24 h of heat stress were greater than

the corresponding numbers in its parents. However, after 72 h of heat stress, the total number of DEGs (868) and the exclusive number of DEGs (180 up-regulated and 203 down-regulated) in the hybrid were smaller than the corresponding numbers in its parents (Fig. 1B). The 24 h and 72 h time points may be key transitional times in the response of II8 to the high temperature conditions. Similarly, after 120 h of heat stress, the exclusive number of DEGs (426 up-regulated and 235 down-regulated) in the hybrid was smaller than the corresponding numbers in its parents.

We detected 1,653 DEGs that were exclusively up-regulated at one or more of the time points and 1,542 of them were over-dominant genes. Among them, 213, 1,315, 327, and 640 were up-regulated at 0 h, 24 h, 72 h, and 120 h, respectively (Table S2). The expression levels of the over-dominant genes were higher in the hybrid than in its parents (*Wei et al., 2009*). The over-dominant effects of genes can combine to promote heterosis (*Shapira & David, 2016*). Using a minimal TPM value of 10 and an absolute value of log2 FC ≥ 10, we detected 73 markedly differentially expressed over-dominant genes that were associated with heat stress responses in the hybrid (Table S3). Approximately 26% of them (19 genes) were annotated as expressed proteins or hypothetical proteins with unknown functions. This suggests that genes associated with heat tolerance in the hybrid are a rich area for further investigations. The expression levels of the small heat shock protein (sHsp) genes MH06g0146900 and MH02g0661300, and the Hsp90 activator gene MH06g0689400 were greatly increased under long-term heat stress. Nine transcription factor (TF) genes encoding TFs in the WRKY, zinc finger, MYB, bHLH, and other families, as well as genes encoding mitogen-activated protein kinase kinase kinase (MAP3K; MH01g0546600) and peroxidase 1 (MH05g0063600) were greatly up-regulated. In addition, genes encoding transposons, retrotransposons, transporters, protein kinases, and protein phosphatases were strongly induced by long-term heat stress. These highly up-regulated genes are involved in a very large proportion of the life processes, indicating that heat tolerance is associated with complicated regulatory networks in the hybrid.

Although short-term heat stress responses have been studied in rice, our results indicate that a large number of genes are positively involved in the heat responses during long-term heat stress.

## Differential gene expression patterns in II8 in response to different durations of heat stress

Details of the heat tolerance mechanisms in the hybrid are not known; therefore, the gene expression levels in II8 flag leaves subjected to long-term heat stress during the flowering stage were studied using a deeper RNA-Seq method. The deeper transcriptome profiles of the hybrid were consistent with the digital gene expression profiling results. The number of up-regulated genes was higher than the number of down-regulated genes (Fig. 2A). Most of the DEGs underwent the highest number of transcriptional changes after 24 h of heat stress. Many of the heat-responsive genes exhibited transient differential expression but reverted to the unstressed levels after 72 h of heat stress. A Venn diagram analysis showed that 586 significantly regulated genes were continuously differentially expressed after 24 h, 72 h, and 120 h of heat stress (Fig. 2B). A hierarchical clustering

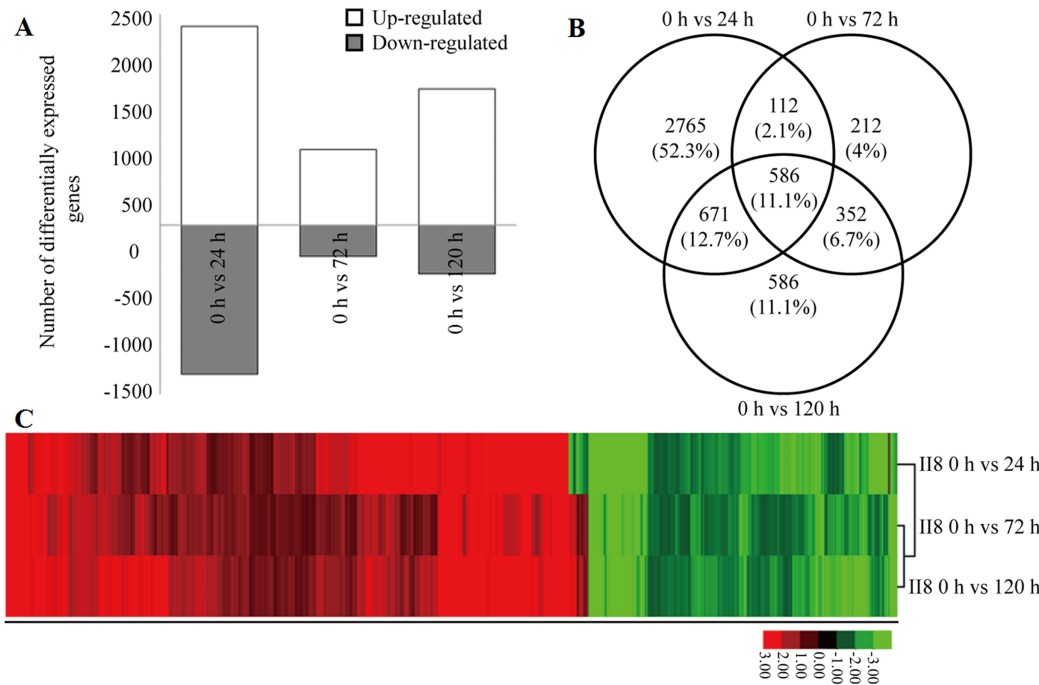

**Figure 2 Differential gene expression in hybrid rice (II8) in response to different durations of heat stress.** (A) Number of differentially expressed genes (DEGs) in II8 after heat stress for 0 h, 24 h, 72 h, and 120 h. (B) Venn diagram of the DEGs in II8 at all the time points. (C) Hierarchical clustering analysis of II8 of 586 significantly regulated DEGs at all the time points.

analysis of these 586 DEGs was performed to identify the co-regulatory pattern after long-term heat stress in the hybrid (Fig. 2C; Table S4). The analysis showed that more genes were continuously up-regulated than continuously down-regulated. The most significantly up-regulated genes were those encoding 3-ketoacyl-CoA synthase 11 (MH06g0569700), heat stress transcription factor B-2c (OsHsfB2c; MH09g0436500), anthocyanidin 5,3-O-glucosyltransferase (MH03g0749700), transposon and retrotransposon proteins (MH06g0215900, MH02g0167600, MH01g0656500, MH07g0247300, MH04g0213900), Hsps (MH03g0171000, MH08g0491200, MH02g0624500), glutathione *S*-transferase (MH01g0791100), MYB (MH12g0137600), histone deacetylase 19 (MH02g0116000), and L-type lectin-domain containing receptor kinase (MH07g0035000), as well as unknown hypothetical proteins (MH09g0263100, MH03g0211100, MH04g0309000).

## Expression pattern changes in II8 with different durations of heat-stress

Because the changes in genome-wide gene expression under heat-stress conditions were complicated, the *k*-means clustering analysis method was used to investigate the expression pattern. The 5,284 DEGs, which were identified in II8 flag leaves at least at one time point, formed 10 clusters (Fig. 3). Cluster A contained DEGs that were continuously down-regulated; Clusters B and C contained DEGs that were continuously

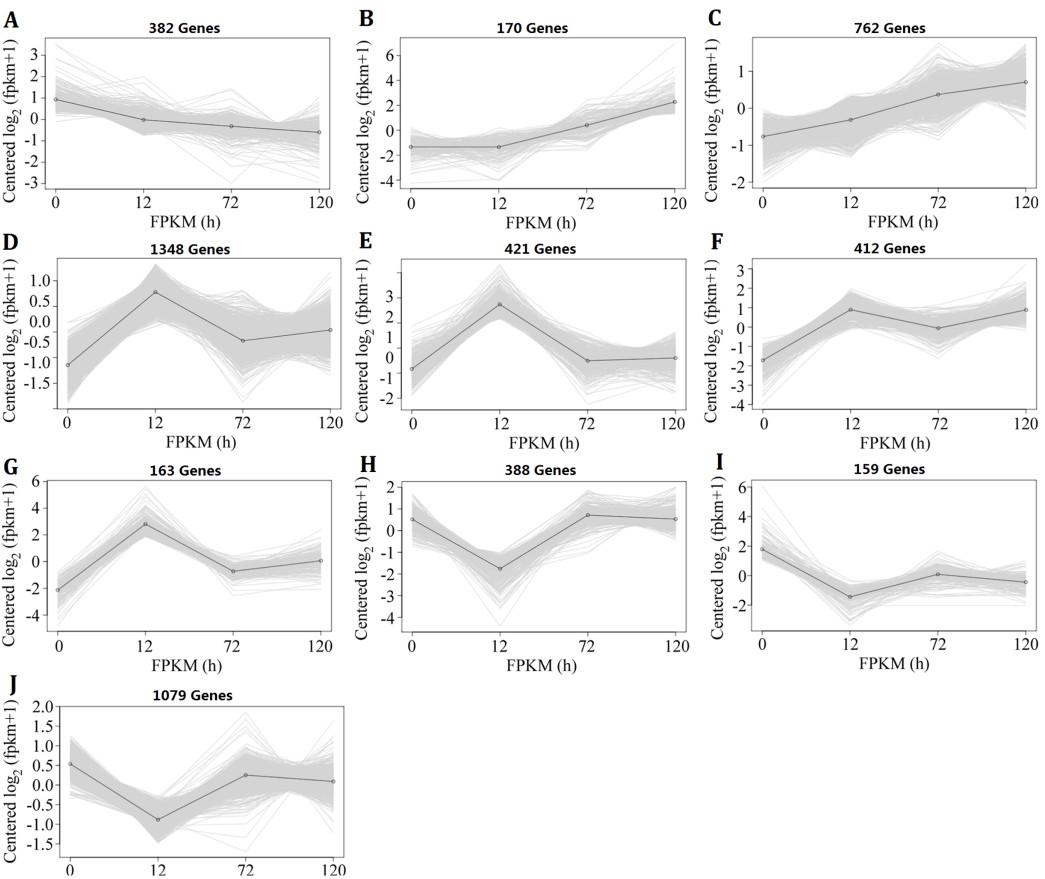

**Figure 3 Expression patterns of the differentially expressed genes in hybrid rice (II8) flag leaves after different durations of heat-stress.** The vertical axis represents the gene expression signal values centered on log2 (fragments per kilobase per million +1), and the horizontal axis represents the four sample times (0 h, 24 h, 72 h, 120 h). (A) Genes in Cluster A were continuously down-regulated. (B and C) Genes in Clusters B and C were continuously up-regulated. (D–G) Genes in Clusters D–G showed early up-regulation and then partial decrease. (H–J) Genes in Clusters H–J showed early down-regulation before reversion.

up-regulated; Clusters D–G contained DEGs that showed early up-regulation and then partial decrease; and Clusters H–J contained DEGs that showed early down-regulation before reversion. The *k*-means clustering results show that the DEGs involved in the heat stress response followed a temporal regulatory pattern.

To determine the processes that were significantly affected by heat stress, a GO enrichment analysis of the DEGs in the early up- and down-regulated clusters, as well as the continuous up- and down-regulated clusters was performed using agriGO (*Tian et al., 2017*). The results showed that most of the up-regulated genes were involved in cell communication (GO:0007154), response to stress (GO:0006950), response to abiotic stimulus (GO:0009628), response to external stimulus (GO:0009605), response to biotic stimulus (GO:0009607), response to endogenous stimulus (GO:0009719), response to stimulus (GO:0050896), metabolic process (GO:0008152) and transcription factor activity (GO:0003700), whereas the continuously down-regulated genes were involved in photosynthesis (GO:0015979) and signal transduction (GO:0007165) (Fig. 4; Table S5).

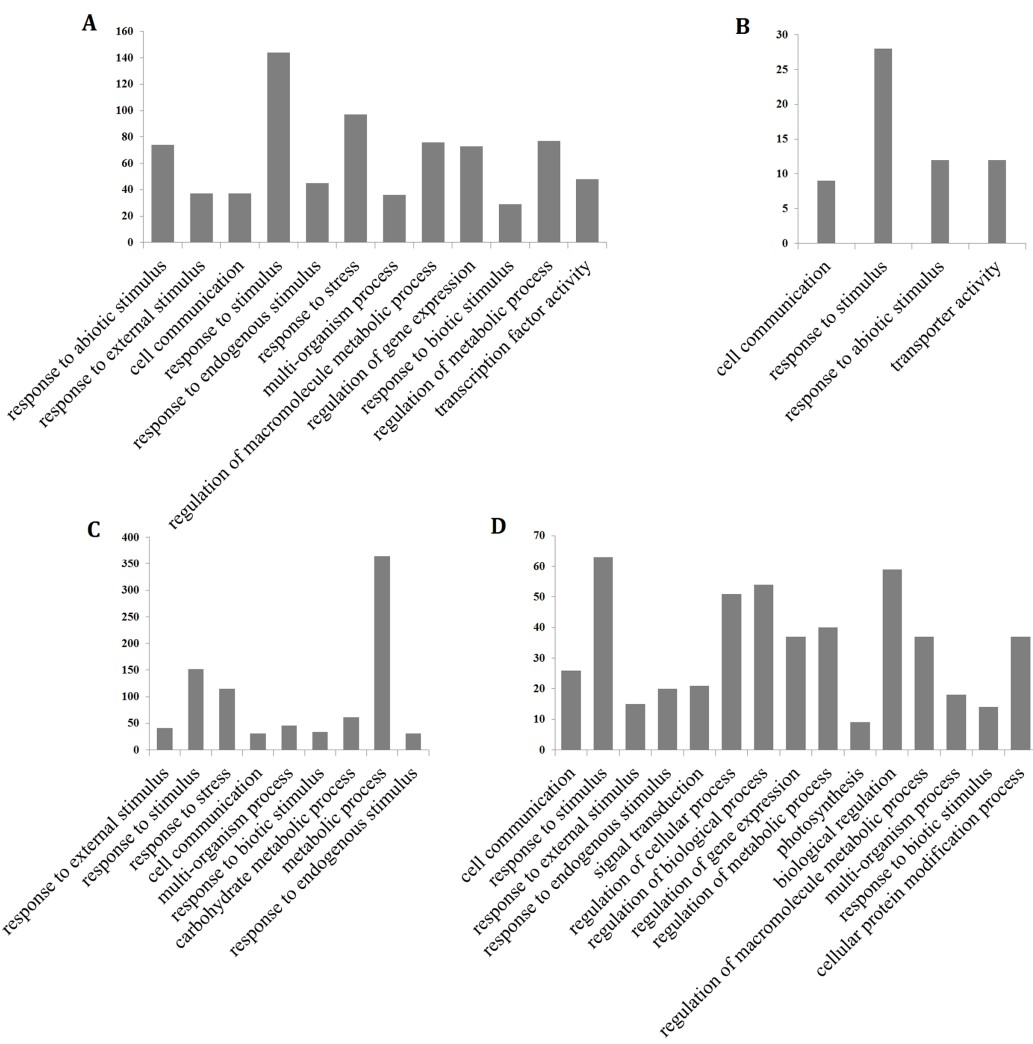

**Figure 4 Gene ontology (GO) enrichment analysis of the hybrid rice (II8) genes in Clusters generated by *k*-means clustering.** (A and B) Distribution of the GO terms assigned to genes in the early up- and down-regulated clusters. (C and D) Distribution of the GO terms assigned to genes in the continuous up- and down-regulated clusters.

The GO terms in the response to stimulus pathway included response to stress, abiotic stimulus, external stimulus, endogenous stimulus, heat, oxidative stress, and biotic stimulus. The genes that were assigned these significant GO terms were modulated by long-term heat stress and are likely to be related to the heat tolerance of the hybrid.

## Identification of TFs in II8 that responded to long-term heat stress

Among the 5,284 DEGs in the hybrid, 160 encoded TFs belonging to 41 TF families, including the WRKY (21 genes), MYB (16 genes), NAC (15 genes), AP2-EREBP (9 genes), bHLH (7 genes), bZIP (7 genes), and HSF (5 genes) families. Additionally, 77 of these TF genes were more highly differentially expressed (absolute value of log2 FC > 2), either up- or down-regulated, at least at one of the time points (Table S6). The heat-responsive

TFs encoded by these genes might work in isolation or act synergistically to regulate the expression levels of a large number of genes.

Among the TF genes, those encoding the WRKY family members were the most numerous (21 genes). WRKY TFs provide enhanced heat and drought tolerance, and function upstream of the HSF–Hsp regulon (*Li et al., 2011*; *Sarkar, Kim & Grover, 2014*). All the WRKY-encoding genes were up-regulated, except MH05g0036100. The transcript levels of MH01g0673800, MH05g0036100, MH05g0538600, and MH08g0380600 were higher in the hybrid than in the parental lines after 120 h of heat stress (Table S7). Members of the HSF family, HsfA4d (MH05g0501600), HsfA2b (MH07g0091300), and HsfB1 (MH09g0368500) were remarkably induced in the hybrid during the long-term heat treatment. The transcript levels of MH05g0501600 and MH09g0368500 were up-regulated early in the hybrid and were higher than their levels in both parents after 24 h of heat stress. Members of the bZIP family, MH06g0591100 and MH07g0509000, were up-regulated in the hybrid on exposure to heat stress and were much higher than in the parental lines after 24 h of heat treatment. The bZIPs have been found to trigger a heat response by unfolded protein, leading to the expression of genes involved in endoplasmic reticulum quality control under heat stress conditions (*Kim, Yamaguchi-Shinozaki & Shinozaki, 2018*; *Lu et al., 2012*; *Takahashi et al., 2012*).

Members of the MYB family, MH07g0545300, MH08g0419700, and MH09g0309700 also more highly expressed in the hybrid than in the parental lines under long-term heat stress. The transcript level of MH09g0309700 was higher in the hybrid than in the parental lines during the 120 h heat treatment. The bHLH family members, MH08g0482900 and MH03g0648500, were up-regulated during the long-term heat stress and were highest in the hybrid after 24 h of heat stress. The NAC family members, MH07g0041200 and MH12g0038200, the FAR1 family member MH01g0445100, the CPP family member MH06g0213800, and the AP2-EREBP family member MH10g0429500 were all remarkably induced in the hybrid during the long-term heat stress. The transcript levels of all these genes were higher in the hybrid than in the parental lines, especially after 24 h of heat stress.

## Validation of gene expression

To validate the RNA-Seq results, eight genes (MH03g0686900, MH03g0260900, MH03g0018700, MH04g0704000, MH07g0536400, MH10g0025400, MH03g0762100, MH05g0501600) were selected randomly for independent validation by qPCR. All eight genes were successfully amplified. As shown in Figs. 5A–5H, the expression trends of the genes determined by qPCR were in good agreement with the trends of the TPM data from the II8 and parental lines. The transcripts levels of the selected genes in II8 differed greatly from those of its parents. A comparison between expression levels of the DEGs obtained by qPCR and the fragments per kilobase per million data obtained by RNA-Seq in II8 flag leaves after different durations of heat stress is shown in Fig. S1. The expression of all the selected genes obtained by qPCR and by RNA-Seq were in good accord.

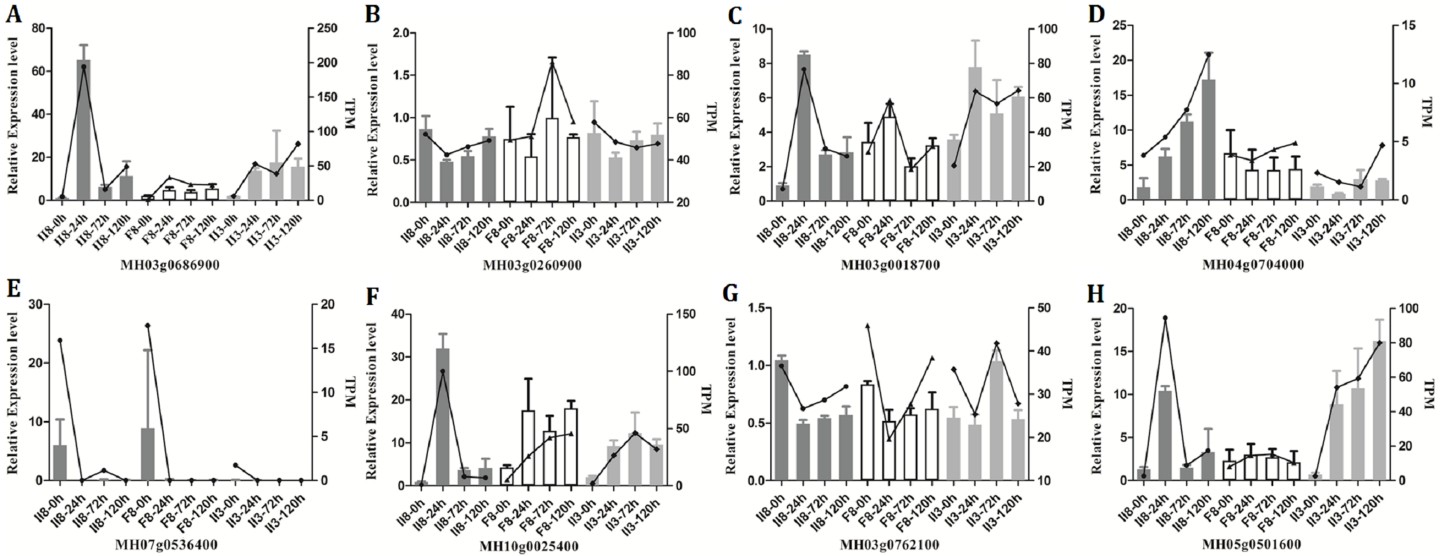

**Figure 5 Quantitative real-time RT-PCR (qPCR) validation of the RNA-Seq data from II8 and the paternal F8 and maternal II3 lines.** The columns indicate the relative expression levels obtained by qPCR and the lines indicate the TPM values obtained by RNA-Seq. Error bars are standard deviations of the mean (*n* = 3) among the three repeats. The internal control genes were MH03g0618400 (ACTIN1), MH01g0139700 (Mpv17), and MH02g0177100 (chloroplast processing peptidase). The genes selected for validation are: (A) MH03g0686900, mitochondrial import inner membrane translocase subunit TIM14-3. (B) MH03g0260900, HUA2-like protein 2. (C) MH03g0018700, Hsp70. (D) MH04g0704000, peroxidase 12. (E) MH07g0536400, peroxidase 2. (F) MH10g0025400, ascorbate-specific transmembrane electron transporter 1. (G) MH03g0762100, OsHsfA1. (H) MH05g0501600, HsfA4d.

## Identification of DAPs under heat-stress conditions

Hybrid flag leaves were heat treated for 0 h and 24 h and used to investigate the heat stress responses at the proteome level. A total of 58 protein spots with FC ≥ 2-fold (*p* < 0.05) in all three replicate samples were considered to be DAPs (Fig. S2). Among them, 35 were unique gene products; seven had increased abundance and 28 had decreased abundance after 24 h hear stress. The 35 unique DAPs were functionally annotated with GO terms (Table 1; Table S8). The most enriched GO terms were response to stimulus, response to oxidative stress, response to abiotic stimulus, and photosynthesis. The DAPs with significantly decreased abundance in response to heat stress were disease resistance protein RGA1 (MH11g0587400), SNF1-related protein kinase regulatory subunit gamma-like PV42a (MH02g0063200), bHLH96 (MH09g0379400), and ferredoxin-dependent glutamate synthase (MH07g0519600). The DAPs with significantly increased abundance in response to heat stress were Hsp70 (MH02g0644700), CPN60 (MH10g0328900), pentatricopeptide repeat-containing protein (MH05g0279300), and calmodulin-binding transcription activator 2 (MH03g0091100). Two disease resistance proteins (MH11g0587400 and MH12g0410900), which are sensors for biotic and abiotic stresses, also were detected among the DAPs.

## Combined analysis of transcriptome and proteome data

The transcriptome and proteome results shared a high rate of coincidence, with 18 of the 35 identified DAPs having the same abundance patterns (increased or decreased) as the corresponding DEG expression levels (up- or down-regulated) (Table 1).

**Table 1 The 35 unique differentially abundant proteins detected in the flag leaves of hybrid rice II YOU 838 after 24 h of heat stress.**

| Spot No.[a] | Gene ID | Functional annotation | Fold change |
|---|---|---|---|
| Response to stimulus | | | |
| 7 | MH02g0458600 | Protein FLUORESCENT IN BLUE LIGHT | 3.74* |
| 9 | MH07g0043700 | Oxygen-evolving enhancer protein 2 | 2.07* |
| 11 | MH12g0410900 | Probable disease resistance protein, leucine Rich repeat family | 3.37 |
| 16 | MH07g0011600 | Glutamate receptor 3.4 | 2.71 |
| 17 | MH02g0063200 | SNF1-related protein kinase regulatory subunit gamma-like PV42a | 7.68 |
| 18 | MH09g0219800 | Phosphoenolpyruvate carboxylase 2 | 2.59 |
| 21 | MH02g0644700 | Heat shock 70 kDa protein | +2.22* |
| 25 | MH11g0587400 | Putative disease resistance protein RGA1 | 8.34 |
| 27 | MH04g0666500 | Glutamine synthetase, chloroplastic | 2.13* |
| 45 | MH11g0166300 | Serpin-Z2B | 2.74 |
| 53 | MH08g0522800 | 70 kDa peptidyl-prolyl isomerase, large rice FK506 binding protein gene rFKBP64 | +2.03* |
| Oxidation-reduction | | | |
| 24 | MH03g0040500 | Glyceraldehyde-3-phosphate dehydrogenase GAPB | 2.04* |
| 52 | MH06g0582000 | Glycine dehydrogenase | +2.10 |
| 54 | MH08g0509600 | Probable lipoxygenase 8 | 2.40 |
| 56 | MH07g0519600 | Ferredoxin-dependent glutamate synthase | 5.08* |
| Photosynthesis | | | |
| 1 | MH07g0421500 | Cytochrome b6-f complex iron-sulfur subunit | 3.40* |
| 2 | MH06g0706000 | ATP-dependent zinc metalloprotease FTSH 1 | +2.08* |
| 6 | MH05g0279300 | Putative PPR | +3.14 |
| 12 | MH07g0124300 | Chloroplast stem-loop binding protein of 41 kDa | 3.23* |
| 15 | MH10g0328900 | Chaperonin 60-2 (CPN60-2), mitochondrial | +2.58* |
| 26 | MH11g0071000 | Fructose-bisphosphate aldolase | 3.46* |
| 42 | MH06g0016000 | Ferredoxin–NADP reductase chloroplast precursor | 3.05* |
| Metabolic process | | | |
| 4 | MH02g0060200 | Tyrosine-sulfated glycopeptide receptor 1 | 2.15 |
| 5 | MH02g0406900 | Probable leucine-rich repeat receptor-like protein kinase | 2.50 |
| 10 | MH04g0024800 | L-type lectin-domain containing receptor kinase IV.2 | 2.68 |
| 20 | MH03g0091100 | Calmodulin-binding transcription activator 2 | +2.06 |
| 40 | MH09g0379400 | Transcription factor bHLH96 | 7.48* |
| 50 | MH10g0280100 | 40S ribosomal protein S28 | 2.17* |
| 55 | MH06g0046100 | Transketolase | 2.10* |
| 3 | MH01g0411700 | MORC family CW-type zinc finger protein 3 | 4.92 |
| 13 | MH05g0540900 | Ketol-acid reductoisomerase | 2.44* |
| 19 | MH11g0317300 | putative reverse transcriptase | 3.12 |
| 33 | MH04g0636300 | Aminomethyltransferase | 2.21* |
| 38 | MH01g0759700 | Ubiquitin-like specific protease 1 | 2.02 |
| 57 | MH07g0520900 | Chromodomain-helicase-DNA-binding protein 1 | 4.83 |

**Notes:**
[a] Spot numbers correspond to the protein spot numbers in Fig. S2. "+" Proteins that showed increased abundance after 24 h of heat stress.
* The 18 proteins that showed the same abundance patterns (increased or decreased) as the corresponding gene expression levels (up- or down-regulated).

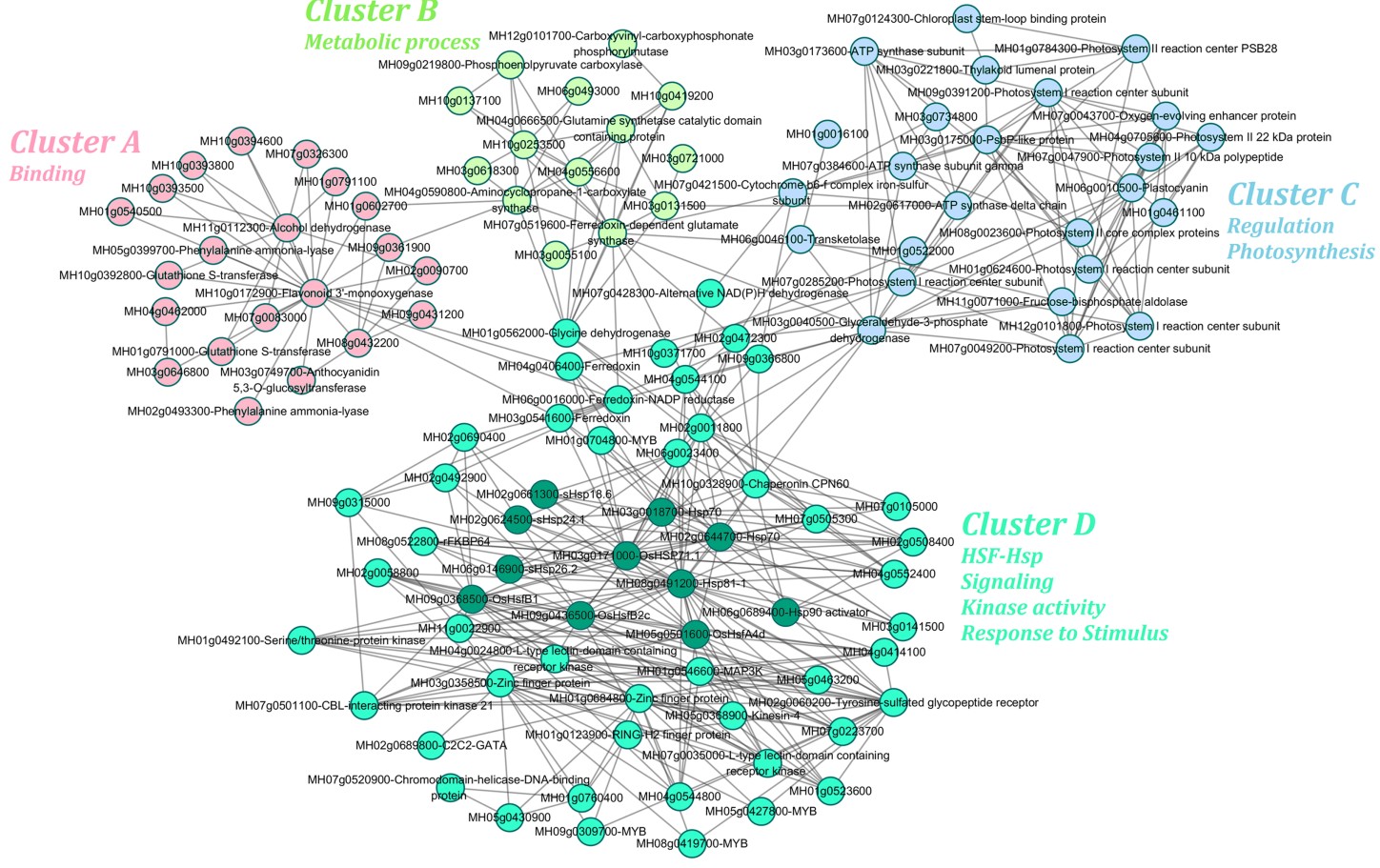

**Figure 6 Co-regulatory network predicted from the differential gene and protein data of II8 flag leaves during long-term heat stress.**
The protein–protein interaction networks were identified using the STRING database. It contained four subclusters; namely, cluster A, cluster B, cluster C, and cluster D. Members of the HSF–Hsp network were dark green nodes in cluster D.

The protein–protein interaction networks of these DEGs/DAPs were predicted using the STRING database and the co-regulatory relationships among them were visualized using Cytoscape (Fig. 6). The heat tolerance co-regulatory network consisted of 117 nodes and 419 edges. Details of the nodes are shown in Table S7. The network contained one large and three small subclusters. The genes in the largest subcluster were most strongly enriched in GO terms related to HSF–Hsp, signaling, kinase activity, and response to stimulus. The genes in the other three subcultures were most strongly enriched in GO terms related to binding; metabolic process; and regulation and photosynthesis.

There were nine hub genes in the network that had more than 10 edges; namely, HSP71.1 (MH03g0171000), Hsp81-1 (MH08g0491200), HsfB1 (MH09g0368500), CPN60 (MH10g0328900), HsfA4d (MH05g0501600), photosystem I reaction center subunit (MH09g0391200), ferredoxin-dependent glutamate synthase (MH07g0519600), alcohol dehydrogenase (MH11g0112300), and flavonoid 3'-monooxygenase (MH10g0172900). More than 100 (25%) of the 419 edges were involved in the HSF–Hsp network, including HsfA4d, HsfB1, HsfB2c (MH09g0436500), Hsp81-1, Hsp70s, and sHsps in the cluster D (Fig. 6). Members of the HSF family, HsfA4d, HsfB1, and HsfB2c were highly connected

with serine/threonine-protein kinase (MH01g0492100) and L-type lectin-domain containing receptor kinase (MH04g0024800, MH07g0035000). Two Hsp70 family genes (MH03g0018700, MH03g0171000) were highly connected with CPN60 and TFs. There were three sHsps (MH02g0661300, MH02g0624500, MH06g0146900) in the co-regulatory network, which indicated that sHsps play key roles in responses to heat stress. *Li & Liu (2019)* reported that sHsps were highly conserved and might have similar functions in mediating the responses of barley plants to heat stress. Members of the Hsp and HSF families were important nodes with abundant connection points in the co-regulatory network. This indicated that the HSF–Hsp network play important roles in the heat tolerance of II8.

While several genes in our co-regulatory network matched genes previously identified in rice seedlings (*Sarkar, Kim & Grover, 2014*), the overall topologies of the rice networks are dissimilar because different sets of input genes were in the co-regulatory analyses. The hierarchical clustering analysis of the 117 node genes revealed that the majority had over-dominant expression patterns in the hybrid, especially after 24 h of heat stress (Fig. S3). These genes likely represent a heat-tolerance strategy in the hybrid during exposure to long-term heat stress.

## DISCUSSION

Rice hybrids have multiple resistance mechanisms to biotic and abiotic stresses, especially resistance to high temperatures, and each of these mechanisms contributes, to some extent, to heterosis (*Feng et al., 2015*; *Hochholdinger & Baldauf, 2018*). In this study, we performed genome-wide transcriptome and proteome analyses of the hybrid rice II8, which is widely cultivated in China, to gain a fundamental understanding of heat tolerance mechanisms of hybrid rice and their contributions to heterosis. Moreover, heat stress induced early response genes that exhibited transient expression but reverted later to unstressed levels. Therefore, the long-term heat response period increased our understanding of the molecular basis of heat tolerance in rice. Our results were not consistent with the observations of *Sarkar, Kim & Grover (2014)* who found that many genes were down-regulated after 1 h of heat stress in 1 month old rice plants. The differences in the results may be because different samples were used. Compared with the parental lines, more II8 genes showed changed expression levels in response to heat stress. This suggested that the hybrid may have a more efficient heat stress response than its parents. The expressed hybrid genes were not only more in number but also more diverse because of the complementation between the parental genomes. These differences in the transcriptomes may explain the superior heat tolerance of the hybrid.

To understand the relationships among the DEGs/DAPs in more detail and to detect possible critical pathways, a co-regulatory analysis was conducted on the genes/proteins identified in the hybrid. In the heat tolerance co-regulatory network, more genes were switched on than switched off after long-term exposure to heat stress. Those increased at the protein and up-regulated at the transcript levels included Hsp70s (MH03g0018700, MH02g0644700) and sHsps (MH03g0686900, MH02g0661300, MH02g0624500). The interaction between Hsp70 and the unfolded protein (its substrate) is regulated by

co-chaperones, including sHsps (*Awad et al., 2008*). Under the same heat stress conditions, the changes in Hsp70 and sHsps transcript levels in II8 were greater than those in the parental lines (Fig. 5). Hsp70 regulates the function HsfA1 through direct interactions, and represses HsfA1 activity (*Hahn et al., 2011*). In the hybrid, Hsp70 was up-regulated and OsHsfA1 (MH03g0762100) was down-regulated over the long-term heat stress period. Unlike the parental lines, in the hybrid, OsHsfA4d (MH05g0501600) and HsfB1 (MH09g0368500) were remarkably induced, especially at 24 h. OsHsfA and OsHsfB sub-families were shown to have important functions in activating the cellular protection system and heat-tolerance in rice (*Zhang et al., 2012b*). HSFs, which activate heat-stress responses, are released from association with, and inhibition by, the Hsp70 and Hsp90 chaperones because the chaperones bind to the misfolded proteins that result from heat stress (*Zhu, 2016*). The Hsps, including sHsp, Hsp90, and Hsp70, and their co-chaperones, function in preventing protein denaturation and maintaining protein homeostasis. The HSF–Hsp regulatory network may be involved in controlling heat stress responses (*Hahn et al., 2011*; *Sarkar, Kim & Grover, 2014*). We hypothesize that the HSF–Hsp network is a main contributor to the heat tolerance of the hybrid.

A wild relative of cultivated *O. sativa* was shown to maintain high levels of the protective proteins Hsp70, Hsp90, and CPN60, as well as a high photosynthetic rate to make it heat tolerant (*Scafaro, Haynes & Atwell, 2010*). In the hybrid, the transcript and protein levels of CPN60 (MH10g0328900) were increased under heat stress, and the rate of increase was greater than in the parental lines. The folding of the ribulose-1,5-bisphosphate carboxylase/oxygenase (Rubisco) large subunit was mediated by the cylindrical chloroplast CPN60 and its co-factor CPN20 (*Wilson & Hayer-Hartl, 2018*). Consistent with the observation by *Gammulla et al. (2010)*, an increase in CPN60 allowed the protein to fold correctly, thereby enabling it to perform its role in the preservation and modification of Rubisco subunits (*Demirevska-Kepova & Feller, 2004*; *Gammulla et al., 2010*). Hsp70 and CPN60 in chloroplasts may sequentially assist in the maturation of newly imported ferredoxin-NADP reductase in an ATP-dependent manner (*Tsugeki & Nishimura, 1993*). The chloroplast genes encoding ferredoxin-NADP reductase (MH06g0016000), cytochrome b6-f complex iron-sulfur subunit (MH07g0421500), and PsbP-like protein 2 (MH03g0175000) were significantly down-regulation after 24 h of heat stress and then reverted back to control levels at 72 h. The transcript levels of these genes were higher in II8 than in the parental lines, which indicated they may play key roles in repairing the damage caused by photo-oxidation after long-term heat treatment (*Cramer, 2019*; *Higuchi-Takeuchi et al., 2011*; *Pawel, Wilson & Gray, 2012*). The chlorophyl content in leaves of creeping bentgrass was found to increase under heat stress (*Liu & Huang, 2000*). Our previous data showed that the chlorophyl content of II8 was higher than it was in the parental lines after heat-stress exposure, and chlorophyl content significant increased after 24 h of heat stress (*Wang et al., 2016b*). Compared with the parental lines, the hybrid can maintain relatively high photosynthetic protein activity levels under the same heat stress conditions. Therefore, these over-dominant genes in the photosynthesis pathway may be related to the higher heat tolerance of the hybrid compared with its parents.

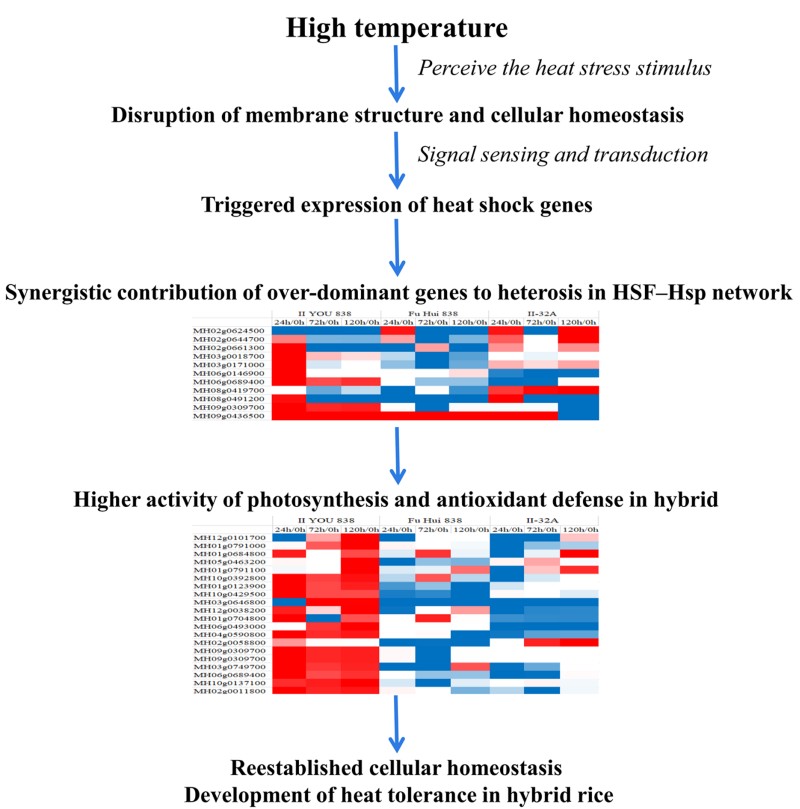

**Figure 7 Schematic illustration of heat tolerance mechanism in the hybrid rice (II8).**

The activity of antioxidant enzymes showed positive correlation with chlorophyl concentration in wheat plant cells under high temperature stress (*Almeselmani et al., 2006*). Genes with a role in antioxidant defense systems, such as the peroxidases (MH04g0704000, MH07g0536400, MH05g0063600), were positively expressed in II8 under heat stress. The transcript levels of peroxidase 12 (MH04g0704000) in II8 were continuously up-regulated and were higher than those of the parental lines under the same heat-stress conditions. Consistent with our previous study, the activity levels of antioxidant enzymes (peroxidase, superoxide dismutase, catalase) in the hybrid were higher than those in the paternal lines under heat stress conditions (*Wang et al., 2016b*). The hybrid was more efficient than the parental lines in scavenging ROS because it had elevated antioxidant transcript and protein levels. The elevated activity of antioxidant enzymes in the hybrid may facilitate the reestablishment of cellular homeostasis to confer resistance to heat stress.

The abundance of calmodulin-binding transcription activator 2 (MH03g0091100) was increased in the hybrid as assessed by the proteomic analysis. Calmodulin is involved in Hsp gene expression through the regulation of HSF activity (*Li et al., 2004*; *Sarkar, Kim & Grover, 2014*). Heat stress also activates MAPKs, which are important for Hsp gene expression and thermotolerance (*Sarkar, Kim & Grover, 2014*). The transcript level of MAP3K (MH01g0546600) in II8 was continuously up-regulated and was higher

than its levels in the parental lines. MAP3K is responsible for MAPK activation in MAPK-signaling pathways related to abiotic stress. MAP3K genes also are up-regulated by heat stress, drought, $H_2O_2$, salt, and cold stresses (*Mittal, Madhyastha & Grover, 2012*; *Sarkar, Kim & Grover, 2014*). Thus, calmodulin and protein kinases play significant roles in signaling during heat stress.

Members of the HSF–Hsp network had over-dominant expression patterns in the hybrid rice compared with their expression patterns in its parents, which may help maintain relatively higher activity levels in photosynthesis and antioxidant defense systems (Fig. 7; Fig. S3). These genes were non-additively expressed under long-term heat stress in the heat tolerance co-regulatory network. The over-dominant mode of inheritance is important to heterosis because a single gene can potentially create the heterotic effect (*Shapira & David, 2016*). Non-additively expressed genes are candidates for contributing to heterosis (*Fujimoto et al., 2018*). These significantly differentially expressed over-dominant genes and non-additive genes may contribute synergistically to heterosis. These genes in the HSF–Hsp network maintained the higher activity of photosynthesis and antioxidant defense systems, that confer the hybrid rice more heat tolerance compared to the parental lines. These genes may contribute to the stronger heat resistance of the hybrid by using a different control mechanism. Thus, the findings of this study will help to advance the understanding of the heterosis of heat stress tolerance in rice.

## CONCLUSIONS

The genome-wide transcriptome and proteome analyses of cultured hybrid rice II8 provides an unbiased methodology to investigate expression patterns of genes and abundances of proteins using digital signals. It is evident that the heat tolerance of hybrid rice is associated with a complicated regulatory network. In the HSF–Hsp network, the over-dominant expressed genes may help to maintain the relatively higher activity levels of Hsps, photosynthesis, and antioxidant defense systems that contribute to the heat tolerance of the hybrid rice. Our results provide insights into the heat tolerance mechanisms of hybrid rice and could be applied to develop new strategies for improving heat resistance in rice.

## ACKNOWLEDGEMENTS

We thank Dr. Wei Li from the University of Michigan for critically reading the manuscript. We thank Lesley Benyon, PhD, and Margaret Biswas, PhD, from Liwen Bianji, Edanz Group China for editing the English text of drafts of this manuscript.

### Funding

This work was supported by IAEA's Coordinated Research Project (No. 16592) and Sichuan Province Science Technology Projects (No. 2018SZ0308). The funders had no role in study design, data collection and analysis, decision to publish, or preparation of the manuscript.

## Grant Disclosures

The following grant information was disclosed by the authors:

IAEA's Coordinated Research Project: 16592.

Sichuan Province Science Technology Projects: 2018SZ0308.

## Competing Interests

The authors declare that they have no competing interests.

## Author Contributions

- Yan Wang analyzed the data, conceived and designed the experiments, performed the experiments, prepared figures and/or tables, authored or reviewed drafts of the paper, and approved the final draft.
- Yang Yu analyzed the data, prepared figures and/or tables, authored or reviewed drafts of the paper, and approved the final draft.
- Min Huang conceived and designed the experiments, performed the experiments, and approved the final draft.
- Peng Gao performed the experiments, authored or reviewed drafts of the paper, and approved the final draft.
- Hao Chen performed the experiments, authored or reviewed drafts of the paper, and approved the final draft.
- Mianxue Liu performed the experiments, prepared figures and/or tables, and approved the final draft.
- Qian Chen performed the experiments, prepared figures and/or tables, and approved the final draft.
- Zhirong Yang analyzed the data, authored or reviewed drafts of the paper, and approved the final draft.
- Qun Sun analyzed the data, conceived and designed the experiments, authored or reviewed drafts of the paper, and approved the final draft.

## Data Availability

The RNA-Seq data are available at NCBI Sequence Read Archive: SRP168528.

## Supplemental Information

Supplemental information for this article can be found online at http://dx.doi.org/10.7717/peerj.8306#supplemental-information.

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
