# Peer review of "Transcriptomic and proteomic profiles of II YOU 838 (Oryza sativa) provide insights into heat stress tolerance in hybrid rice"

_PeerJ, doi:10.7717/peerj.8306_

## Round 0.1 · original submission · Major Revisions

You manuscript requires additional work. First of all, please reconsider changing the title to better reflect the content of your paper. Second, the introduction should be re-worked, so the readers have a complete picture of the heat-stress response mechanism in rice, and about importance of this problem.

Quality of written English should be improved, typos eliminated and grammar checked by a native speaker.

Presentation and organization of results also should be improved. Please think about a logical way to tell your story.

Materials and methods section needs additional works, please add more details.

Reviewer 1 ·

Basic reporting

More background about the mechanism of heat-tolerance in rice.

Experimental design

No comment.

Validity of the findings

No comment.

Additional comments

Wang et al described a genome-wide transcriptome and proteome analysis of hybrid rice. Their work identified the potential role of a complicated regulatory network in heat tolerance of II YOU 838, including HSF-Hsp, photosynthesis and oxidation-reduction. The experiments are well designed and the manuscript is well organized. However, there are several points I feel concerned.

Major points
1) Proteome data were used in combined analysis with transcriptome data, which also provided key evidences. Hence, the title should be improved.
2) The introduction section is too weak. I failed to gain a whole picture of the mechanism of heat-tolerance in rice or even other plants from this section.
3) The authors described the results in an illogical order. For example, DEGs encoding TFs were described, followed by those encoding PALs, GSTs, and Hsps and so on in line 185-197. Why?
4) They speculated that HSF-Hsp might be important for heat tolerance in II8, and explained a lot of their evidences. However, too little information was given in the figures.
5) Some references are missed.


Minor points
1) Line 35: ‘help to’.
2) Line 44-46: References are needed.
3) Line 60-62: References are needed.
4) Line 70: Confused. I think they want to say the overlook of early responses of heat shock in previous studies.
5) Line 83: ‘Sichuan’.
6) Line 113: Please check the reference.
7) Line 129: ’ACTIN1’.
8) Line 150-157: The authors described the flag leaf phenotype and identified II8 to be more tolerance to heat stress. Is there any physiological evidence?
9) Line 176: What does they mean by over-dominant genes?
10) Line 184: I feel it is not so interesting.
11) Line 260 and 262: What does the WRKY family mean? The whole family in rice? Or DEGs belonging to WRKY?
12) Line 260-262: Which WRKY? Is it included in the DEGs?
13) Line 277: ‘than in’.
14) Line 290-298: Which RNA samples were used? Are they identical to those for RNA-seq? Why?
15) Line 353-356: Those two sentences should be appeared in Introduction section.
16) Fig 3: The cluster name and number of DEGs of each cluster should be given in the figure. The order of each plot should be identical to the appearance order in the text.
17) Fig 4: The word size is too small.

·

Basic reporting

The current MS is on comparative transcriptomics and proteomics analysis of heat resistant hybrid rice and their parent lines. Given the current and alarming problem of global warming, the aim of this study is very significant.
1. The title of this MS is too conclusive while the results and discussion are not directed towards that.
2. There are few typos (e.g. line 48: nighttime should be night time), please go through the text again.
3. The introduction is not very detailed and you are citing almost all of the references more than 5 years old. While there is a number of new reports on rice and increasing temperature.

Experimental design

1. Please provide in details about planting materials and growth conditions. From the current methods, it’s hard to understand how you started your planting material. Is it from plates to the growth chamber or directly sown in field and later transferred to growth chamber?
2. “Heading stage plants were transferred to growth chamber” Please also mention in terms of days after sowing/plating.
3. At which stage tissue were collected? Please also mention in terms of days after transfer to the growth chamber.
4. We know that most of the genes are regulated by circadian rhythm, so it is also important to mention the time of tissue collection.
5. The color resolution of Fig 1A is very poor. It will be good if you provide pics from 0 hr to 120 hrs, this will help us to understand the dynamic changes under heat stress.
6. Table 1 and 2, should go-to supplement. You may pick the top 10 candidates from each table and make one table instead of long two different tables.
7. Starting from the title of this MS and several places in the text, ‘mechanisms’ is the word of choice, but I didn’t see any single experiment/ data which is directed towards this. At least at the end, a model figure may be helpful which is very much missing.

Validity of the findings

Based on only RNA-seq and proteomics analysis, it is very hard to come to any conclusive picture of the problem. It is now highly recommended to verify the results of transcriptome/proteome by at least picking one/two candidates and go for further characterization.

---

## Round 0.2 · accepted · Accept

Thank you very much for investing significant time to improve your manuscript. I will recommend acceptance of your paper for publication.

Reviewer 1 ·

Basic reporting

no comment

Experimental design

no comment

Validity of the findings

no comment

Additional comments

This is a resubmission with significant changes according to the reviewers' suggestions. After reading the revision, I am satisfied by the changes the authors made in this revision. The revision addressed all of concerns and, as I can see from the responses to the reviewers' comments, also addressed most, if not all, concerns and questions raised by other reviewers. I think this revision has significantly improved the quality of the paper.

·

Basic reporting

Based on earlier suggestion and review, authors improved this MS very significantly and responded to all the queries.

Experimental design

Improved and upto the mark now.

Validity of the findings

This revised MS is improved. All the findings were discussed in details with relevant citation.